# Tiny Guides, Big Impact: Focus on the Opportunities and Challenges of miR-Based Treatments for ARDS

**DOI:** 10.3390/ijms25052812

**Published:** 2024-02-28

**Authors:** Chirag M. Vaswani, Julia Simone, Jacqueline L. Pavelick, Xiao Wu, Greaton W. Tan, Amin M. Ektesabi, Sahil Gupta, James N. Tsoporis, Claudia C. dos Santos

**Affiliations:** 1Department of Physiology, Temerty Faculty of Medicine, University of Toronto, Toronto, ON M5S 1A8, Canada; c.vaswani@mail.utoronto.ca (C.M.V.); greaton.tan@mail.utoronto.ca (G.W.T.); 2Keenan Research Centre for Biomedical Science, St. Michael’s Hospital, University of Toronto, Toronto, ON M5B 1W8, Canada; maxxiao.wu@mail.utoronto.ca (X.W.); amin.ektesabi@mail.utoronto.ca (A.M.E.); jim.tsoporis@alum.utoronto.ca (J.N.T.); 3Department of Medicine, McMaster University, Hamilton, ON L8V 5C2, Canada; 4Institute of Medical Sciences, Temerty Faculty of Medicine, University of Toronto, Toronto, ON M5S 1A8, Canada; jacqueline.pavelick@mail.utoronto.ca; 5Faculty of Medicine, School of Medicine, The University of Queensland, Herston, QLD 4006, Australia; sahil.gupta@uq.net.au; 6Laboratory Medicine and Pathobiology, Temerty Faculty of Medicine, University of Toronto, Toronto, ON M5S 1A8, Canada; 7Interdepartmental Division of Critical Care, St. Michael’s Hospital, University of Toronto, Toronto, ON M5B 1W8, Canada

**Keywords:** ARDS, gene therapy, miRNA, nanotechnology

## Abstract

Acute Respiratory Distress Syndrome (ARDS) is characterized by lung inflammation and increased membrane permeability, which represents the leading cause of mortality in ICUs. Mechanical ventilation strategies are at the forefront of supportive approaches for ARDS. Recently, an increasing understanding of RNA biology, function, and regulation, as well as the success of RNA vaccines, has spurred enthusiasm for the emergence of novel RNA-based therapeutics. The most common types of RNA seen in development are silencing (si)RNAs, antisense oligonucleotide therapy (ASO), and messenger (m)RNAs that collectively account for 80% of the RNA therapeutics pipeline. These three RNA platforms are the most mature, with approved products and demonstrated commercial success. Most recently, miRNAs have emerged as pivotal regulators of gene expression. Their dysregulation in various clinical conditions offers insights into ARDS pathogenesis and offers the innovative possibility of using microRNAs as targeted therapy. This review synthesizes the current state of the literature to contextualize the therapeutic potential of miRNA modulation. It considers the potential for miR-based therapeutics as a nuanced approach that incorporates the complexity of ARDS pathophysiology and the multifaceted nature of miRNA interactions.

## 1. Introduction

There are over 60 known causes of Acute Respiratory Distress Syndrome (ARDS) [1,2], including pneumonia (e.g., infection by bacteria [3,4], viruses [5,6,7,8], and fungi [9]), systemic diseases (e.g., sepsis) [10,11], severe trauma [12], multiple transfusions [13], chemotherapy [14], toxic inhalations [15], pancreatitis [16], and chemical/biological warfare [17]—all of which may require advanced respiratory support such as supplemental oxygen, mechanical ventilation (MV), or extracorporeal membrane oxygenation (ECMO) [18]. 

Irrespective of the cause, all pathways of acute lung injury (ALI)—the biological process underlying ARDS—converge on shared histological and molecular features, including flooding of the lung airspaces with edema fluid [19], activation of immune cells and secretion of inflammatory mediators causing immune dysregulation [20], destruction of lung tissue, and degradation of surfactant. Increased dead space (perfusion of non-aerated alveoli) and decreased pulmonary compliance (stiffening of the lungs) result in impaired gas exchange; thus, the need for respiratory mechanical support [20,21,22]. When injury in the lung reaches a critical threshold, inflammation spills over into the circulation (loss of pulmonary compartmentability), to affect other organs, leading to multi-organ failure (such as heart, kidney, brain, and liver failure) and ultimately death [23]. Currently, there are no specific treatments for ARDS; nonetheless, efforts are being made to tease out “treatable traits”, i.e., sets of specific problems that can be managed by an individualized treatment programme [24].

In this new era of genomic medicine, RNA-based therapies are particularly attractive, especially when transient regulation of gene expression profiles may be enough to redirect innate immune responses without permanently altering the genetic makeup of the cell. MicroRNAs (miRs) are small RNAs with a length of 20–24 nucleotides that control gene expression post-transcriptionally [25,26,27]. miRs work as gene silencers by complementary base-pairing to the 3′ untranslated regions (UTRs) of mRNAs [26,28,29]; miRs can then destabilize and degrade the target mRNA or suppress its translation [30,31]. The requirement for base-pairing by way of nucleotides at positions 2–8 of the mature miRNA termed the “seed sequence” makes them highly specific [32]. Herein lies the attraction of miRs as basic therapeutic agents: delivery of miR mimics or inhibitors can be used to target therapeutically relevant miRs [33,34]. Because miRs occur naturally and can bind to multiple genes with variable strength and specificity by altering the number of nucleotides that are complementary to the seed sequence (the more/wider the coverage the higher the binding specificity), miR-based therapies can be both biologically effective, specific, and relatively safe [31,35]. 

## 2. Identifying Therapeutically Relevant microRNAs

miRs seem to be involved in all cellular processes; while detecting differentially expressed miRs associated with particular conditions, distinguishing causally and therapeutically relevant miRNAs from epiphenomenal changes is challenging. The approach taken by various groups, including our own, is to utilize advanced algorithms to effectively analyze large and complex networks of relationships between biomedical entities, and to embrace data-driven decision-making in target assessment and prioritization to promote innovation by using digital discovery platforms to generate innovative hypotheses about potential therapeutic targets that can be tested in vitro and in vivo. 

The discovery approach in our lab relies upon system perturbation and identification of attractor states in high dimensional data analysis [36]. In the case of our ARDS studies, we perturbed the system by treating septic mice with mesenchymal stromal cells (MSCs) [37,38,39,40]. MSCs have been shown to effectively prevent lung injury and organ dysfunction by reducing inflammation and enhancing bacterial clearance [39,41,42,43,44]. We investigated the transcriptome and microRNAome of cases versus controls treated with MSCs versus placebo to identify mRNA:miR pairs regulated in both cases and controls, as well as placebo vs. MSC in five major organs affected by sepsis (lungs, liver, kidney, spleen, and heart). We combined gene set enrichment analysis with co-regulation of mRNAs and miRs to identify possible therapeutic targets on the assumption that, while one regulated miR may be interesting, cumulative co-regulation of an miR (or a group of related miRs) and its known putative target (or group of targets) in all conditions (e.g., sepsis) across more than one organ (e.g., all 5 organs we looked at) denoted a highly causal relationship with the condition or trait of interest, allowing us to filter out epiphenomenal changes [37]. Candidate mRNA:miR pairs that emerged from this analysis were ranked and prioritized for future empiric validation using strict criteria, including: (i) statistical significance of changes (standard deviation amongst replicates), (ii) magnitude of change (effect size), (iii) sequence homology between mice and humans (to facilitate future translation to humans), (iv) pathways enrichment that related to disease of interest (biological plausibility), and (v) the ability to experimentally test a novel in silico hypothesis—to measure and evaluate an effect on putative biotargets and function in vitro and in vivo. 

In our first paper, we identified miR-193b-5p and its predicted target of the tight junctional protein occludin (Ocln). While the occludin gene was known to carry single nucleotide polymorphisms associated with ARDS, the lack of phenotype in Ocln-deficient mice had dissuaded investigators from pursuing its role in the pathophysiology of ARDS [37,39,40]. To empirically validate our new candidate biotargets, we used gain and loss of function studies, transgenic mice, and molecular biology (luciferase 3′UTR constructs) to demonstrate the binding and relative contributions of miR-193b-5p to occludin regulation in sepsis-induced ARDS. 

Importantly, identified biomarkers may be associated with but not causatively related to the disease pathophysiology. For instance, Zhu et al. used miR profiles and logistic regression analyses to identify the miR markers associated with ARDS [45]. However, this does not imply a cause-effect relationship between the dysregulation of the miRNA and disease pathogenesis. Experimental validation is fundamental to verifying in silico predicted relationships and biological effects. In a study of Staphylococcal Enterotoxin B (SEB)-induced lung injury, Rao et al. examined significant differentially expressed miRNAs in vehicle- and SEB-exposed mice and found that miR-155 was the most highly expressed microRNA in the lung [46]. In subsequent supporting experiments, the team showed that miR-155 regulates IFN-γ production by targeting Socs1, indicating its high therapeutic potential in treating SEB-induced ALI [46]. While computational analysis could detect microRNAs with statistically significant changes in expression level, their biological relevance in the development or the treatment of ARDS needs to be validated with appropriate cell or animal models. Experimental validation often involves manual manipulation of a single microRNA, facilitating the further characterization of its specific therapeutic or pathogenetic role and the magnitude of its impact in ARDS. 

In linking specific miRs with ‘treatable traits’, García-Hidalgo et al. utilized statistical models to investigate the relationship between miRs and pulmonary function, providing insight into the potential molecular pathways for SARS-CoV-2-induced ARDS [47]. Lung diffusing capacity for carbon monoxide (D_LCO_) and total severity score (TSS) were used as outcome variables in this study. The association between them and miRs was determined by random forest (RF) for multivariate analyses and generalized additive models (GAMs) for univariate analyses, adjusted for variables such as age, sex, and previous pulmonary disease, to identify miRs that were thought to significantly contribute to pulmonary sequelae [47]. However, while building such models and determining their statistical significance, we also need to incorporate existing knowledge about the biological nature of the disease and miRs. This strategy is a cost-effective way to narrow down the pool of candidate miRs, which can be further investigated and validated using wet-bench experiments.

Research on miRs employs multiple computational methods and experimental validation to elucidate the significance of miRs in gene expression alteration and disease pathogenesis, thereby presenting a comprehensive overview of the molecular changes involved. In a study of miR-384-5p in LPS-induced ALI, information from TargetScan, miRanda, and miRBase was combined to determine the target of miR-384-5p, confirmed by luciferase assay [48]. Another study conducted by Li et al. identified common differentially expressed genes among COVID-19, ARDS, and sepsis patients, followed by gene ontology (GO) analysis, Kyoto Encyclopedia of Genes and Genomes (KEGG) pathways, and enrichment analysis using DAVID [49]. Protein-protein interaction (PPI) networks were further constructed in STRING, which integrates known and predicted PPIs. Cytohubba, a plugin in Cytoscape, was used to extract the hub gene in the PPI network to make further predictions [49]. Tarbase and mirTarbase determined potential regulatory miRs for differentially expressed genes [49]. Li et al. also investigated potential drug molecules using Enrichr with the Drug Signatures database (DSigDB) and examined gene-disease association via NetworkAnalyst with the DisGeNET database [49]. 

With many computational methods available, selecting the most suitable approach for a given study can be challenging. While each gene set collection contains different genes and pathway annotations, which can be determined based on the specific research focus, comparing different gene set analysis methods is more complicated. Using semi-synthetic simulation data, Mathur et al. compared the statistical properties of four gene set analysis methods, including Gene Set Enrichment Analysis (GSEA), Significance Analysis of Function and Expression (SAFE), sigPathway, and Correlation Adjusted Mean RAnk (CAMERA) [50]. Overall, sigPathway is the most powerful method, while GSEA can have similar performance by adjusting the default q-value calculation [50]. In a comparative study conducted by Bayerlová et al., one overexpression analysis (ORA) method (Fisher’s exact), two functional class scoring (FCS) methods (Wilcoxon rank sum and Kolmogorov–Smirnov), and four pathway topology-based (PT) methods (SPIA, CePa ORA, CePa FCS, and PathNet) were examined [51]. Using simulation and benchmarking data, PT methods only exceeded ORA/FCS methods with non-overlapping pathways [51]. Development of future computational software that will allow for streamlining the data-mining and enrichment analysis workflow will be instrumental in advancing miR-based research.

## 3. miRNA-Based Treatment for ARDS?

Overexpressing miRs can be achieved with miR mimics, synthetic double-stranded RNA oligonucleotides, as shown in Table 1. Mimics interact with endogenous miR processing machinery and are sorted into RNA-induced silencing complexes (RISCs) to interact with their target transcript. Due to this interaction, minimal chemical modification of the oligonucleotides is required to optimize efficacy. Modifying the passenger strand with 2′-methylated nucleosides ensures that the correct strand is incorporated into RISCs [52]. Ensuring that passenger strands are not also incorporated can reduce the side effects of miR therapy by increasing target specificity [52]. RNases also readily degrade endogenous miR, posing a limiting factor to miRNA-based treatments [53,54].

Alternatively, the effects of an endogenous miR can be mitigated by miR inhibitors [55,56], miR sponges, and molecular inhibitors that impede miR–mRNA interaction. miR inhibitors, chemically modified nucleic acids also called antagomirs or antisense oligonucleotides (ASO), work by binding to miRs via complementary base-pairing to form an RNA duplex that will then be degraded by endogenous RNAse H. In addition to pre-clinical studies, anti-miR therapies have been evaluated in Phase I and II clinical trials, as shown in Table 2. RGLS4326, a chemically modified anti-miR-17, is in a phase I clinical trial to reduce cyst growth in polycystic kidney disease [57]. A recent dose escalation phase I study of locked nucleic acid (LNA) interfering with miR-221 was shown to be safe and possibly efficacious for treating refractory multiple myeloma and advanced solid tumours (NCT04811898) [58]. 

In parallel, miR sponges are RNA constructs containing multiple miR binding sites that ‘soak-up’ the miR of interest to decrease its availability to bind with its mRNA targets [59]. This method of inhibition was exemplified by one study that demonstrated an effective decrease in miR using a circular RNA containing two binding sites for miR-550a in human breast cancer cells [60]. While miR sponges have proven helpful in determining miR function, plasmid safety and potential for off-target interaction often limit their clinical use as therapies. 

Previous groups have shown the therapeutic efficacy of miRNA inhibitors or mimics in various models, including cancer [61,62,63], influenza virus [38], hepatitis C virus [64,65], chronic inflammatory diseases [66,67], cardiac injury [68], fibrosis [69,70], and metabolic diseases [71,72].

**Table 1 ijms-25-02812-t001:** Summary of reviewed articles on the use of miRNAs in models of lung injury.

	miRNA	Direct Target	Pathway	Target Organ/Cell	Expression When Therapeutic	Carrier	Route	Source
Adverse	miR-155	SOCS1	NF-κB	Macrophages	Downregulated	Exosome	Injected intravenously	[73]
	miR-193b-5p	Occludin	Unknown	BEAS2b, HPMECs and mouse lungs	Downregulated	HiPerfect reagent	Injected intratracheally	[38]
	miR-762	NFIX	miR-762/NFIX	A549 and HEK293T	Downregulated	Lentivirus	Injected intranasally	[74]
Protective	miR-27a-5p	VAV3	Unknown	Mouse lungs	Downregulated	HiPerfect reagent	Injected intratracheally	[40]
	miR-34b-5p	PGRN	Unknown	Lung homogenates	Downregulated	None	Injected intravenously	[55]
	miR-221	SOCS1	NF-κB	RAW264.7 cells and mouse lungs	Downregulated	None	Injected intravenously	[56]
	miR-126	SPRED1	RAF/ERK	HUVEC and mouse lungs	Upregulated	Exosome	Injected intravenously	[75]
	miR-384-5p	Beclin-1	Possibly Autophagy (Not fully known)	Alveolar macrophages and Mouslungs	Upregulated	Exosome	Injected intravenously and intratracheally	[48]
	miR-371b-5p	PTEN	PI3K/Akt	Human primary ATIICs and mouse lungs	Upregulated	Exosome	Cell experiment	[76]
	miR-125b-5p	Keap1/Nrf2/GPX4	Keap1/Nrf2/GPX4	PMVEC	Upregulated	Lipofectamine	Cell experiment	[77]
	miR-223	PARP-1	NF-κB/AP-1	Mouse lungs	Upregulated	Neutral lipid emulsion (Lipid nanoparticle)	Injected intratracheally	[78]
	miR-23b-3p	FGF2	NF-κB	Mouse lungs and BMSC	Upregulated	Lentivirus	Injected intratracheally	[79]
	miR-127	CD64	IgG Fcγ Receptor I	RAW264.7	Upregulated	Lentivirus	Cell experiment	[80]
	miR-200c/b	ZEB1/2	p38 MAPK and TGF-β/smad3 (Unknown)	RLE-6TN (rate alveolar cell) and mouse lung	Upregulated	Lentivirus	Injected intratracheally	[81]
	miR-506	p65	NF-κB	Mouse lung	Upregulated	Lentivirus	Injected endotracheally	[82]
	miR-193b-3p	β-catenin	Wnt/β-catenin	A549 and Mouse lung	Upregulated	Adenovirus	Injected intratracheally	[83]
	miR-454	CXCL12	CXCL12/CXCR4	Mouse lung	Upregulated	Adeno-associated virus	Tail vein injection	[84]
	miR-4262	Bcl-2	Unknown	Mouse lung	Upregulated	Adeno-associated virus	Tail vein injection	[85]

**Table 2 ijms-25-02812-t002:** Summary of miRNAs used in clinical trials.

Drug	miRNA	Drug Type	Carrier	Phase	ClinicalTrials.gov Identifier	Illness	Source
RGLS4326	Anti-miR-17	Locked nucleic acid (LNA) inhibitor	Unknown	Phase I	NCT04536688	Autosomal dominant polycystic kidney disease	[57]
LNA-i-miR-221	miR-221	Inhibitor	Unknown	Phase I	NCT04811898	Refractory advanced cancer	[58]
MRX34	miR-34a	Mimic	Liposomal nanoparticle	Phase I (Terminated)	NCT01829971	Refractory advanced cancer	[86]

## 4. What Are the Advantages of miRNA Therapy for Complex Acute Conditions?

One of the main advantages of miR-based therapies is that they modulate a naturally occurring regulatory system. Endogenous mechanisms have evolved for the cell to manage exogenous RNA as a host defence from pathogens; by manipulating naturally occurring regulatory systems, miR-based therapies may be less likely to trigger this same host response [87]. 

Second, the ability of miRs to target multiple transcripts in different pathways may induce a broader response [88,89]. This is important given the complexity and the redundancy of the innate immune response and the heterogeneity of responses to an acute insult [90]. With this intricacy of the inflammatory response in mind, it has become clear that targeting a ‘single’ mediator is unlikely to effectively change outcomes in the critically ill—likely because of the complex and intricate complementation and compensation circumventing any beneficial effect of single mediator therapy [91]. MiRs target entire networks of genes, which may enable us to stimulate or dampen the activity of a broad network effectively [92,93]. 

Thirdly, given the severe, disseminated, but transient nature of immune dysregulation during injury, infection, and inflammation, approaches that rely on a “hit-and-run” effect (exposure-response therapy) might be more effective by working quickly to mitigate innate immune dysfunction and reconstitute homeostasis [94]. 

Fourthly, it may be possible to tailor therapy to specific miRs with both biomarker and biotarget function (theragnostic). Given delivery vehicles can be manufactured on demand and be used to deliver various miR-based therapeutics simultaneously, combination therapy would be relatively feasible and safe. 

Finally, by creating a library of miR-based therapies tailored to address specific treatable traits, designing single or combination miR-based delivery vesicles for individualized therapy may be feasible, making personalized off-the-shelf therapeutics for complex acute care syndromes possible. 

The perceived advantage of broad effect may also be the “Achilles heel” of miR-based therapeutics. An unfortunate example of this is the clinical study with miR-34a mimic MRX34, in which the trial was prematurely terminated due to severe immune-related side effects that resulted in the death of four patients [86,95]. MRX34 was tested as a tumour suppressor therapy and was delivered using a pH-dependent delivery strategy to target the low-pH environment of tumours [95]. However, in clinical testing, miR-34a was found in white blood cells, and animal model testing identified miR-34a uptake in the spleen [96]. It is now understood that miR-34a plays a role in tumour suppression and immune cell signalling [97]. Fortunately, as our understanding of miR-based therapeutics improves, so does our ability to make therapies safer. Recently, a press release accompanied the results of a clinical study utilizing a first-in-class, fully modified version of miR-34a with outstanding stability, activity, and anti-tumour efficacy. The fully modified microRNA-34a rendered the miR nearly invisible to the immune system. They further reduced toxicity by conjugating the miR to a targeting ligand that has high affinity and specificity for a receptor upregulated by the target cells; in the case of miR-34a, this was conjugated to the folate receptor that is overexpressed in cancer cells [98]. Better designs moving forward are expected to reduce toxicity, increase specificity, facilitate correct intracellular distribution, and increase RNA stability.

## 5. How Can miRNA Be Modified to Optimize Delivery?

As oligonucleotides, miR-based therapies must overcome degradation and avoid immune stimulation. This can be done by modifying the 2′-OH of the ribose sugar backbone, such as a 2′-*O*-methyl group, which does not affect miRNA mimic function [99,100,101]. Moreover, it has been found that single-stranded RNA interference (RNAi) therapies are more likely to activate the immune system than double-stranded RNA, encouraging the transfection of double-stranded RNA to be processed by the endogenous miR synthesis pathway to reduce immune stimulation [102]. Pre-clinical studies of various inflammatory lung pathologies have successfully demonstrated positive outcomes with miR mimic administration in vivo with these advancements [82,103,104].

Additionally, chemically modifying miRs allows for direct manipulation of pharmacokinetic and pharmacodynamic properties, which can be used to optimize dosing and therapeutic ranges. The dosing of miR-based therapies significantly affects target interaction and off-target effects [105]. The effects of an miRNA-based therapy may be investigated through miR knockout transgenic mice lines and delivering the miR inhibitor as a control. Subsequently, RNA can be profiled using transcriptomics. While these strategies address the effects that one miR may have on a variety of target transcripts, it is also true that one transcript can be acted upon by more than one miR [106]. Therefore, it can be challenging to predict the extent to which a given dose of miR-based therapy will induce target gene silencing. It is also essential to recognize that miR-based treatments rely on uptake by endogenous RISCs, which can lead to saturation and subsequent interference with other endogenous miR pathways [107,108]. This is also seen with siRNA- and shRNA-based therapies [107,108]. 

Knowledge gaps in effective miR-based therapy development mainly involve strategies to predict appropriate dosing, tissue-specific targeting, and immune system activation [88,89]. These concerns can, to some extent, be addressed by the chosen vector for miR delivery.

## 6. What Vectors Can Be Used for miRNA Delivery?

As discussed, miRs are prone to extracellular degradation, off-target complementary binding, and immune stimulation. For this reason, the vector used to deliver miR therapeutics can significantly impact efficacy, dosing, and toxicity. Vectors used for miR delivery include lipid nanoparticles, exosomes, and viral vectors. 

## 7. Lipid Nanoparticles

Lipid nanoparticles (LNPs) are spherical membranes composed of ionizable lipids, which are positively charged at low pH and neutral at physiological pH environments. LNP lipid composition promotes plasma membrane interaction and facilitates systemic cellular uptake via receptor-mediated endocytosis. The use of lipid nanoparticles in mRNA SARS-CoV-2 vaccines demonstrates the potential of LNPs as vectors for nucleic acid delivery. Moreover, extensive research and now widespread use of LNP technology can significantly reduce the burden of seeking pre-clinical approval, as demonstrated by the expedited roll-out of the Pfizer-BioNTech (BNT162b2) and Moderna (mRNA-1273) SARS-CoV-2 vaccines. Additionally, ONPATTRO (Patisiran) became the first FDA-approved siRNA-based therapy in 2018, in which siRNA-containing LNP vectors are used to treat transthyretin-induced amyloidosis [109]. Not only are there FDA-approved therapeutic agents that deliver mRNA or siRNA using an LNP vector, but there are also pre-clinical studies using LNPs to carry miRs. For example, one study demonstrated the efficacy of in vivo LNP delivery of a tumour suppressor miR as a cancer therapy in mice [110].

All FDA-approved LNP particles contain an ionizable cationic lipid, helper lipids, cholesterol, and polyethylene glycol (PEG)-lipid conjugates [111,112,113], each contributing to LNP size, stability, cellular uptake, and other important factors that impact vector efficacy [114]. A commonly used lipid vector is the liposome. Liposomes can be cationic; however, the positive charge can be toxic in high doses, promote inflammation, and interact with endogenous negatively charged proteins. Trang et al. observed that a neutral lipid emulsion carrying miR-34a and let-7 in a mouse non-small cell lung cancer model disseminated more evenly to different tissues [115]. In contrast, cationic particles tended to collect in the liver [115]. Ionizable cationic lipids permit pH-dependent positive charge, meaning the LNP would remain positive in acidic environments but neutral otherwise. This allows a neutral charge in the blood, reducing cytotoxicity, but a positive charge in acidic environments, facilitating nucleic acid interaction and endosomal membrane fusion for cargo release.

PEG-lipid components improve the half-life of circulating LNPs [116], but a high concentration of PEG-lipid in the LNP can impede endocytosis [117]. For this reason, the concentration of PEG-lipid in the LNP is minimized as much as possible; Semple et al. aimed to optimize cationic lipids for siRNA delivery and noted vital findings such as optimal pKa constants for LNPs, and ideal PEG-to-ionizable lipid component ratios [118]. PEG-lipids that can diffuse out of the LNP have also been explored, demonstrating a lengthened circulation time [119,120,121]. In contrast, permanently PEGylated LNPs are more immunogenic, resulting in rapid clearance, an antibody-driven reaction, and reduced potency upon subsequent exposure [122,123]. Besin et al. recently demonstrated that LNP administration in two-week intervals or with wash-out periods between injections significantly reduced blood clearance [124]. Notably, this experiment was done in an animal model with no previous exposure to PEG-lipids; human beings, however, are more likely to have prior exposure to PEG-containing products such as vaccines and common medications. Nonetheless, PEG-lipids that can diffuse from the LNP offer a strategy to reduce immunogenicity and improve half-life without requiring a more intensive administration schedule. 

Empty LNPs (eLNPs), or vehicle-only controls, have been shown to promote pro-inflammatory IL-6 production and T-follicular helper cell activation in mice [125]. Further studies identified eLNP-mediated maturation and activation of dendritic cells and monocytes, leading to the secretion of pro-inflammatory cytokines and activation of cell-signalling pathways [126]. This presents an important consideration when choosing negative controls for LNP vector testing in pre-clinical and clinical research. 

## 8. Extracellular Vesicles

Extracellular vesicles (EVs) are endogenous cell-to-cell communicators that have also been explored as carriers of oligonucleotide-based therapies. Exosomes are a subset of EVs, and can emerge from nearly all cell types. Exosomes are formed via invagination of endosomal membranes, creating multivesicular bodies (MVBs) [127]. They are then taken up through fusion of the MVBs and target cell plasma membranes. Exosomes are composed of lipids and surface proteins from their origin cells. This cell-specific lipid and protein composition can affect targeting and distribution. Transporter proteins such as CD13, and fusion proteins including flotillin and annexin are key proteins that can differ depending on the cell source and dictate target cell delivery [128]. 

Endogenous exosomes carry a variety of cargo, including miR [129]. One such example includes miR-317b-5p, carried in exosomes derived from alveolar progenitor type II cells (ATIIC) in the lungs, which was found to promote re-epithelialization of injured alveoli and modify ATIIC proliferation in an in vitro model of alveolar injury [76]. As natural miR carriers, exosomes present significant potential for miR therapy delivery due to minimal immunogenicity and protection from RNase degradation [130]. Moreover, the receptor-mediated targeting of exosomes allows for localization to a particular cell type, minimizing off-target interactions. Examples of exosome-mediated miRNA delivery to the lungs are discussed in more detail in a subsequent section.

## 9. Viral Vectors 

Viruses are another group of delivery vectors for miR-based therapeutics [131]. To carry genetic material, the critical viral genes are replaced with the desired transgene, including miRs [132]. Types of viral vectors used in laboratory or clinical settings to carry miRs include adenoviruses (Ad), adeno-associated viruses (AAV), herpes viruses, poxviruses, retroviruses, and lentiviruses [76,81,131]. One study used a retrovirus vector to deliver NF-κB p65 siRNA in a sepsis-induced ALI mouse model [133]. In the case of retrovirus-based vectors, the main limitation is that replicating their cargo and delivering the gene therapy requires cell proliferation, which is not always present with lung cells [134]. Alternatively, lentiviral vectors can infect a range of dividing and non-dividing cells, integrating into the host genome stably without affecting their normal function [135]. Lentiviral vectors contain plasmids that fall into two parts: packaging system and transfer vector [135]. The packaging system is required for viral particle formation and infectivity, while the transfer vector is required for mobilizing the viral genome [135]. The transfer vector contains the transgene and cis-acting sequences for RNA production and packaging [135]. Lentiviral vectors have been used to deliver miR-127 to reduce CD64 expression and lung inflammation [80]. Lentivirus-carrying miR-23b-3p or miR-762 inhibitors have been used to promote lung repair and reduce lung injury characterized by alveolar epithelial cell destruction, increased blood-air barrier permeability, and non-cardiogenic pulmonary edema [74,79]. Despite some degree of success in preclinical models, safety concerns—the most pressing of which is the activation of protooncogenes—remain [136].

Ad and AAV-derived vectors are the most studied and commonly used viral vectors [137]. AAV vectors have a protein shell that protects a single-stranded genome with three genes, producing nine gene products essential for genome replication, packaging, and cell binding [138]. AAV depends on co-infection with other viruses for replication, and hundreds of unique strains across different species are known [138]. Recombinant AAV (rAAV), which lacks viral DNA, can be engineered to deliver an oligonucleotide cargo to the nucleus of a cell [138]. The advantages of rAAVs include low pathogenicity, low immunogenicity, low risk for insertional effects, and the ability to infect both dividing and non-dividing cells in a broad spectrum of tissues [135,137]. One study delivered miR-454 using an AAV to inhibit the translation of C-X-C motif chemokine 1 (CXCL1) in human lung epithelial cells, resulting in a reduction in the permeability index and the production of inflammatory cytokines [84]. AAVs have also been used to successfully deliver miR-4262 to decrease pulmonary cell apoptosis and the severity of ALI [85]. While AAVs can successfully deliver miRNAs to the lungs, their transfection of specific cell types and tissues may be low, and they have limited packaging capacity [135,137].

Compared to AAV, Ad has a larger genome and contains double-stranded linear DNA [132]. Ad comprises a capsid surrounding the DNA core and the genome [139]. They are less favourable than AAVs because Ad vectors are highly immunogenic [140]. However, Ad vectors can still deliver miRs to the lungs, as seen in a study that delivered miR-193b-3p in vivo to suppress influenza viral infection in mice [83]. Overall, viral vectors can be used for miR delivery, but many potential health risks of viral vectors need to be further investigated.

## 10. Delivery to Lungs

Most clinically tested miR therapeutic candidates are delivered through intramuscular or intravenous injection [89]. While systemic administration has been shown to provide treatment to the lungs, direct pulmonary routes of administration may mitigate off-target effects and increase bioavailability at the target site. 

Pre-clinical studies that have explored direct pulmonary delivery of RNAi therapeutics tend to use intratracheal or intranasal administration rather than inhalation, likely due to ease of use in animal models. For example, Courboulin et al. delivered miR-204a via intratracheal nebulization to the lungs and demonstrated improved pulmonary hypertension with decreased vascular remodelling [141]. Additionally, intratracheal administration of miR-223 mimic in a model of ventilation-induced lung injury (VILI) successfully ameliorated ALI [78]. Importantly, ALN-RSV01 was administered intranasally to treat respiratory syncytial virus (RSV) infection in lung transplant patients, becoming the first siRNA therapy to use direct lung administration in clinical trials [142].

While intratracheal and intranasal delivery methods still have clinical relevance, intranasal administration is hindered by filtration in the human nasal cavity, and the invasive nature of intratracheal administration limits generalizability. Though less common, animal models using inhalation have demonstrated the efficacy of oligonucleotide therapeutics [143,144]. Notably, Eluforsenm, an antisense oligonucleotide delivered via oro-tracheal (OT) administration that targets the mutated mRNA of the cystic fibrosis transmembrane conductance regulator (CFTR) protein, completed Phase Ib clinical trial (NCT02532764) in 2017 [145]. Additionally, MRT5005, which delivers fully functional CFTR mRNA via nebulization, has completed Phase I and II clinical trials, and in 2020, the FDA granted Fast Track and Rare Paediatric Disease designations, expediting the approval process [146].

Inhalation delivery of RNAi therapeutics requires careful consideration of vector composition. While naked siRNA and mRNA have been successfully delivered to the lungs, it is essential to recognize that this is only effective for specific cell types that reside in the lung [147,148,149]. To demonstrate this, Ng et al. delivered naked siRNA intratracheally to the lung tissue of mice, noting significant siRNA effects in lung epithelial cells, dendritic cells, and alveolar macrophages [150]. Importantly, while the siRNA was not delivered in a vector, it was chemically modified to ensure stability and mitigate immunogenicity [150]. Moreover, while some propose that lung surfactant plays a role, the specific process of how the naked siRNA entered lung cells remains unknown [151,152]. Notwithstanding, using vectors improves RNAi delivery to the lungs compared to naked RNA [153,154,155].

## 11. Lipid-Based Vectors for miR Delivery to Lungs

In direct pulmonary administration, a limitation to lipid-based vectors is structural stability due to fusion with lung surfactant, resulting in premature RNA release [156]. Specific composition profiles of LNPs have been shown to improve stability, as is demonstrated by MRT5005 clinical studies transfecting mRNA via nebulization [146].

Advancements in direct lung delivery of mRNA via nebulization, in which the medication is aerosolized and inhaled, have shown promise [157]. One study compared different LNP compositions and molar ratios to define optimal LNP makeup for low-dose delivery of mRNA to the lungs via nebulization in mice, which differed from optimization for systemic delivery [158]. This method was tested in an H1N1 mouse model, in which mRNA for a neutralizing antibody was loaded on the LNP vector [158]. PEG molarity, density, and structure significantly impacted delivery performance, as well as the inclusion of cationic lipids [158]. 

Nebulized formulations of LNPs can have advantages for treatments for pulmonary targeting. Compared to naked oligonucleotides, LNPs greatly decrease RNase-mediated degradation and ensure a significant proportion of the therapeutic dose or active therapeutic ingredient is delivered to the lungs, thus exposing affected tissues to greater therapeutic concentrations of the product [159]. For instance, inhaling 100–200 µg of salbutamol was found to have the same effect as an oral dose that was 20 times larger, ranging from 2 to 4 mg. Encasing drug payloads in LNPs can prolong lung retention [160]. How the airway cells take up the drugs, either through endocytosis or the drug’s separation from the LNP carrier, aids in the regulated and extended release of the active therapeutic ingredient. As a result, the therapeutic effect of the payload can last longer between dose administrations [161,162]. This allows for achieving the desired therapeutic outcome with smaller doses, potentially reducing the risk of administering excessive amounts and inducing systemic side effects. LNP formulations incorporating bio-similar phospholipids, cationic or ionizable lipids, and cholesterol are less likely to trigger immune responses as components such as phosphatidylcholine and disaturated phosphatidylcholine are found in human lung surfactant fluid [160,161,162,163]. This promotes physiological compatibility, which minimizes clearance by alveolar macrophages and can increase the therapeutic treatment window [162,164]. 

Clinical case reports suggest intravenous PEG-lipids administered intravenously trigger anti-PEG IgG and IgM antibody production [165,166]. Secondary lymphoid organs can trigger memory formation in the adaptive immune system, rendering subsequent treatments with PEGylated LNPs less effective [165,166,167]. The respiratory tract, however, has fewer IgG and IgM antibodies and is primarily protected by the innate immune system [165]. Therefore, administering drugs through the inhaled route, particularly in LNP form, can reduce the likelihood of inciting the adaptive immune response compared to systemic administration or using inhalation of naked oligonucleotides [160,162,164,167,168].

Furthermore, the deposition of inhaled substances in the respiratory tract depends on various factors, with the aerodynamic diameter (AD) of the nebulized formulations playing a key role in where it will end up along the respiratory conducting and non-conducting zones [169,170]. LNPs with an AD between 1 and 5 μm are more likely to settle in the lower airways, reaching even the farthest bronchioles and alveoli due to gravitational sedimentation. This is conducive for deeper penetration into lung tissue [170,171]. LNPs larger than 5 μm in AD are most likely to get “stuck” in the upper airways due to inertial impaction and may even enter the digestive tract instead of the respiratory system [164,169,170,171,172,173].

While direct lung delivery of LNPs is promising, miR-containing LNPs can also be localized to the lungs following systemic administration. Recently, selective organ-targeting (SORT) LNPs have significantly improved the intravenous delivery of RNA-based therapeutics to the lungs. SORT allows organ-specific targeting by modifying classic LNP compositions to include an additional lipid component coined a “SORT molecule”. SORT molecules are designed to alter the internal charge of the LNPs to allow for extra-hepatic selectivity; to formulate lung-targeting LNPs, the cationic lipid SORT molecule 1,2-dioleoyl-3-trimethylammonium-propane (DOTAP) is added at a 50% molar ratio of total lipid [174]. 

Alternatively, localization may be achieved using LNP modifications targeting lung tissue following systemic administration. Parhiz et al. achieved this via IV injection of mRNA-loaded LNPs with monoclonal antibodies specific to PECAM-1, the vascular cell adhesion molecule in the endothelial cells of the lungs [175]. They reported 200 times higher lung localization with significant specificity to endothelial cells. LNPs have shown promise in delivering oligonucleotide-based therapies and specific lung tissue targeting. 

Furthermore, intravenous delivery may be a more reasonable and strategic option for nanoparticle systems, such as those that hitchhike red blood cells (RBC) or situations where the lungs may be filled with protein exudates and hyperactive immune cells, which may act as a barrier that prevents LNP entry into host cells [176].

Lung-specific targeting can also be achieved by coating the LNP surface with targeting ligands [152]. Coating hybrid nanoparticles with endogenous surfactant protein B (SP-B) has significantly improved intracellular siRNA delivery [177]. Conjugation of LNPs with macromolecules, such as GALA peptide (a 30 amino acid synthetic peptide with a glutamic acid-alanine-leucine-alanine repeat) [178], or antibodies, such as anti-plasmalemma vesicle-associated protein [179], has demonstrated preferential delivery of LNPs to the lungs after intravenous administration. Importantly, LNPs interact with proteins in the blood and incorporate them onto their surfaces, forming a targeting ligand coat of serum proteins called a protein corona. This protein corona can cause unexpected tissue localization of LNPs even when formulated to be organ-specific with the SORT methodology as certain host serum proteins also contribute to organ targeting. Qiu et al. demonstrated that by altering the linker structure in the lipidoid tails of SORT LNPs, the composition of the serum proteins that are incorporated into the protein corona can be manipulated to shift the LNPs towards lung-specific delivery [180].

## 12. Delivering Exosomes to the Lungs

Careful selection of exosome subtypes and genetic modification of exosome membrane proteins can facilitate targeted biodistribution [181]. Exosomes used in pre-clinical studies have successfully delivered miR to the lungs in lung injury disease models. For example, intravenous delivery of miR-155 results in macrophage activation and subsequent lung injury in mice [73]. Additionally, intravenously administered bone marrow mesenchymal stem cell (BMSC)-derived exosomes containing miR-384-5p improved pulmonary vascular permeability in a rat model of LPS-induced ALI [48]. Moreover, Shen et al. delivered exosomes transfected with miR-125b-5p mimic intravenously in mice, demonstrating reduced pulmonary microvascular endothelial cell ferroptosis in a septic lung injury model [77]. In another study, exosomes derived from endothelial progenitor cells (EPCs) were shown to shuttle miR-126 to endothelial cells and mitigate lung injury after LPS [182]. Subsequently, miR-126 mimics were transfected into EPCs to produce miR-126-containing exosomes, which had a protective effect in LPS-induced lung injury [75]. 

Alternatively, direct delivery of extracellular vesicles can also target therapy to the lungs. Bandeira et al. demonstrated that intratracheally delivered extracellular vesicles obtained from MSCs mitigated fibrosis and reduced inflammation in the lungs of a silicosis mouse model [183]. 

These examples demonstrate success in exosome-mediated miRs delivery in pre-clinical studies of lung injury. However, the focus of the current literature remains on identifying endogenous exosomes secreted in disease pathologies and identifying their cargo, which may include miRNA. As such, in vivo studies exploring therapeutic miRNA delivered by exosomes are not as ubiquitous as other delivery vectors discussed, namely LNPs and viral vectors. 

## 13. A Few Words about Silencing-(si)RNA-Based Therapies 

RNAi involves double-stranded RNA that silences mRNA via complementary binding, resulting in post-transcriptional gene regulation. RNAi includes miR and siRNA, which are both generated by double-stranded RNA processed by Dicer, and incorporated into RISC with Ago2 interaction. miR and siRNA also share many physical properties as they are both oligonucleotides. These connections between miR and siRNA implicate similar advantages and barriers in therapeutic delivery, including extracellular degradation of nucleic acids and strategies of lung localization.

One key differentiating factor between siRNA and miR is that siRNAs are specific to a single mRNA transcript while miRs can act on multiple transcripts in different cellular pathways. This difference is explained by the full complementarity of siRNA to the target mRNA, contrasting the complementarity of miR to shared target seed sequences. Another difference between siRNA and miR is that while siRNA can only be used to produce a silencing effect of one target mRNA, miR offers both the option to inhibit or increase the function of an existing endogenous miR. As such, miRs can be studied both as a therapeutic agent and a drug target.

Furthermore, miRs can be used as disease biomarkers as endogenous host entities. A shared barrier of RNAi includes the reliance on endogenous RISC uptake, which can result in the saturation of this machinery and subsequently impede endogenous miRNA pathways. The off-target effects in the perturbations of endogenous miR via RISC saturation can be more unpredictable, making dosing in RNAi-based therapies an essential factor in risk assessment.

Interestingly, siRNA-based treatments may have ‘miR-like effects’ when an siRNA binds the 3′UTR of an off-target mRNA [184,185]. This can often be avoided with more careful siRNA design, such as preventing complementarity to miRNA seed regions already identified in miRNA databases. Another proposed method of reducing off-target siRNA effects is siRNA pooling, in which multiple siRNAs that silence the same transcript are used together at low concentrations [185]. In this case, the adverse-effect profile of a single siRNA is diluted due to lower dosing, but target mRNA silencing is still adequately achieved [185].

Another challenge to siRNA design is immune activation [186,187]. The mechanism of siRNA-mediated immune activation can be either sequence-independent, via TLR3, or sequence-dependent, via TLR7 on dendritic cells and TLR8 on monocytes [188,189]. The chemical modification of immunogenic sequences on the siRNA or use of siRNA delivery vectors can mitigate these effects. The mechanism of sequence-specific immune stimulation is not fully understood, requiring careful and extensive testing for possible immunogenic effects of siRNA therapies.

Despite these challenges, pre-clinical and clinical trials have demonstrated effective siRNA delivery with significant therapeutic benefits in inflammatory lung disease. Several pre-clinical studies have tested siRNAs against TNF in models of inflammatory lung injury and demonstrated therapeutic benefit [190,191,192]. SiRNA with other transcript targets have also exhibited protective effects against lung injury. Many of these studies made use of LNP-based delivery systems. In one pre-clinical study, siRNA inhibited the enzyme that phosphorylates paxillin, a protein involved in lung epithelial permeability [193]. 

## 14. Concluding Remarks

ARDS is a serious pathology that has major impacts on the health care system due to its frequency in ICU, the need for escalation to mechanical ventilation, and high mortality. There are no targeted pharmacologic treatments available for ARDS, and it is predominantly managed supportively with ventilation strategies. A growing foundation of knowledge on miR has encouraged the exploration of potential therapeutic uses. First, the advancements in identifying key miR in major physiologic and disease pathways offer potential therapeutic targets. Subsequently, strategies to modulate specific miR expression can include miR mimics that are agonistic to an endogenous miR or inhibitors that prevent the interaction of a specific endogenous miR with its mRNA targets. These modalities provide the opposite effect, inhibiting miR that has been implicated in disease or amplifying others that have been observed in beneficial host response. Advances to miR-based therapies include modifications to the miR themselves as well as to their vectors of delivery. Specific to advancements pertinent to ARDS treatment, different strategies have proven successful for lung localization. This is especially exciting as it allows for increased drug concentration in affected lung tissue and minimizes off-target effects. Compared to other developing therapeutic options, miR has the major benefit of potentially altering entire gene pathways, which can be especially powerful in complex processes such as ARDS. Other cell-free methods, such as antibodies and siRNAs, have shown promise in pathologies with single gene or protein malfunction; however, the need to identify a single target whose inhibition would significantly alter the disease course in ARDS proves to be a major limitation. Additionally, despite a more targeted mechanism of action, off-target adverse events and harmful immune stimulation are still seen in these options. As such, miR continues to show promise for therapeutic use in lung disease. Moreover, the gross effect of miR modification via inhibition or agonism may be the most effective strategy in tackling enormous pathways such as ARDS.

## Data Availability

No new data were created or analyzed in this study. Data sharing is not applicable to this article.

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
