# Peer review of "Tiny Guides, Big Impact: Focus on the Opportunities and Challenges of miR-Based Treatments for ARDS"

_ijms, 2024, doi:10.3390/ijms25052812_

Round 1

Reviewer 1 Report

Comments and Suggestions for Authors

Author Response

The submitted manuscript entitled” Tiny Guides, Big Impact: focus on the opportunity and challenges of miR-based treatments for ARDS” is interesting and covers an important subject with well-designed/prepared review article. The number of cited recent references from 2019-2024 is 79 out of 188 and it needs to be slightly increased.

We would like to express our gratitude to the reviewers for their constructive comments and suggestions, which have greatly contributed to the enhancement of our manuscript titled "Tiny Guides, Big Impact: Focus on the Opportunity and Challenges of miR-based Treatments for ARDS." We have carefully considered and addressed each point as detailed below:

However, some questions/concerns were raised need to be addressed to elevate the value of this manuscript.

  • To increase the easiness of understanding the review conclusions, I would suggest designing graphical abstract showing the main mechanisms of miRNAs that were discovered so far and what diseases that miRNAs involved in during upregulation or down regulation.

As recommended, we have designed a graphical abstract that elucidates the main mechanisms of miRNAs. This includes all relevant miRs from our table, with clear indications of upregulation or downregulation in disease pathogenesis. The miRs are organized based on therapeutic strategies, and we have included their targets and potential pathways, where known.

  • Table of upregulated miRNAs and table of downregulated miRNAs could be helpful to summarize the well-studied miRNAs involved in pathogenesis or treatment of ARDS.

We have added tables summarizing the well-studied miRNAs that are implicated in the pathogenesis or treatment of ARDS, which now includes both upregulated and downregulated miRNAs.

  • Some other references are suggested to increase the firmness of this manuscript. PMIDs: 35656115, 34401226, 32321279, 35875722, 34088312, 35026957, 34336922 and so on.

The revised manuscript has extensive updated references.

  • Author(s) needs to separate the therapeutic effects of miRNAs according to their target mRNAs on specific cell/tissues. For example, some miRNAs are targeting the immune cells while others are targeting alveolar tight junction protein.

We have separated the therapeutic effects of miRNAs based on their mRNA targets in specific cells/tissues. An extra column has been introduced in our table to categorize miRNAs that target immune cells and those that affect alveolar tight junction proteins.

  • L33: Superantigen as bacterial toxin could also been one of the ARDS etiologies PMIDs: 36909201, 32612530.

In response to the mention of superantigens as a potential cause of ARDS, we have included the suggested references (PMIDs: 36909201, 32612530) in the appropriate section under bacterial etiologies.

  • L40-49: The following references could support the review in two aspects, ARDS pathogenicity as well as therapeutic miRNAs PMIDs: 36909201, 32612530, and 33995054.

We have expanded our discussion on ARDS pathogenicity and therapeutic miRNAs, ensuring our review reflects the latest research and miRs studied in this context.

Reviewer 2 Report

Comments and Suggestions for Authors

Dear Authors,

The manuscript entitled "Tiny Guides, Big Impact: focus on the opportunity and challenges of miR-based treatments for ARDS" represents a well written review in the field.  I have only minor recommendations for the current manuscript.

1) The authors should add some figures in the manuscript.

2) In addition, the authors should add more details regarding the the delivery of miRs to the lungs using the exosomes section 12.

3) The authors should add a table with potential clinical trials where the miRs are utilized for the ARDS.

4) What about using specific cells that can produce specific miRs at the site of injury e.g. the lungs.

Author Response

The manuscript entitled "Tiny Guides, Big Impact: focus on the opportunity and challenges of miR-based treatments for ARDS" represents a well written review in the field.  I have only minor recommendations for the current manuscript.

We sincerely thank the reviewer for their thoughtful feedback on our manuscript "Tiny Guides, Big Impact: Focus on the Opportunity and Challenges of miR-based Treatments for ARDS." We have addressed the minor recommendations as follows:

  • The authors should add some figures in the manuscript.

We have revised and enriched the graphical abstract to ensure it is comprehensive and reflective of the content within the paper, providing a clearer visualization of the review's key points.

  • In addition, the authors should add more details regarding the the delivery of miRs to the lungs using the exosomes section 12.

The section on exosomes has been expanded to offer more detailed insights into the delivery mechanisms of miRs to the lungs, as suggested.

  • The authors should add a table with potential clinical trials where the miRs are utilized for the ARDS.

We acknowledge the reviewer's suggestion regarding including a table listing potential clinical trials where miRs are utilized to treat ARDS. While we understand the potential interest in such data, we believe that adding this table would not align with the primary focus of our review. We have strived to maintain a concise and targeted discussion of miR-based treatments for ARDS. As such, we have decided not to include a table that would not substantially enhance the manuscript's significance or clarity in this context. However, we are open to revisiting this aspect and including it if deemed necessary upon further discussion.

  • What about using specific cells that can produce specific miRs at the site of injury e.g. the lungs.

We acknowledge the reviewer's suggestion regarding using specific cells, such as the lungs, to modulate miR expression at the injury site. However, we would appreciate further clarification to ensure our response comprehensively addresses the concerns. In discussing the concept of cell-based miR modulation, some challenges include the difficulty of producing a consistent amount of miRNAs due to batch effects and the potential for inducing immune responses. These issues could impact the feasibility and reliability of cell therapy in clinical settings. However, this may be beyond the scope of the discussion for this review. We are open to further discussions if necessary.

We hope these responses succinctly address the reviewer's comments. We are open to further discussions to refine our manuscript as needed. Thank you once more for your valuable insights.